# Species-Specific Response to Climate Change: Evident through Retrospective Analysis Using Tree Ring Data

Prem Raj Neupane [1,2,]*[ID], Archana Gauli [1,2], Rajendra KC [3][ID], Buddi Sagar Poudel [3] and Michael Köhl [1][ID]

1   Institute of Wood Science—World Forestry & Center for Earth System Research & Sustainability (CEN), University of Hamburg, Leuschner Strasse 91, D-21031 Hamburg, Germany
2   Friends of Nature, Kathmandu 44618, Nepal
3   Ministry of Forests and Environment, Singh Durbar, Kathmandu 44600, Nepal
*   Correspondence: prem.raj.neupane@uni-hamburg.de

**Abstract:** It is likely that changing monsoon patterns and changes in other climatic parameters will have an impact on forests. Tree growth and biomass may respond differently across the different forest recovery contexts after the disturbance regimes. It is essential to understand the response of different tree species in order to comprehend their ability to adapt to climate change. An enhanced understanding of how tree species dynamics change with a substantial shift in climate attributes is crucial to develop adaptive forest management strategies. Advances in the statistical application of tree ring data results in more reliable dating with the high accuracy and precision of any of the paleo-records and robust and long-term reconstructions of key indices such as temperature and precipitation. In this study, we analyzed how different species inhabiting together respond to changes in climatic variables using dendroclimatic analysis. We assessed the growth performance of *Abies pindrow*, *Pinus wallichiana*, and *Tsuga dumosa* in the temperate region of Nepal. The climate sensitivity of the species was analyzed using bootstrap correlation analysis and the climate-growth relationship over time was assessed using the moving correlation function. Tree ring growth of *Abies pindrow* is stimulated by higher June temperatures and higher March precipitation. This positive relationship is consistent and stationary over time. However, in the other two species, both response function and moving correlation analysis showed that the relationship between climate and growth is inconsistent and changes over time.

**Keywords:** dendroclimatology; tree growth; *Abies pindrow*; *Pinus wallichiana*; *Tsuga dumosa*

## 1. Introduction

Scholarly and political interest in damages and irreversible losses resulting from climate change-induced extreme weather events and slow-onset processes in terrestrial ecosystems has grown substantially over the last few decades [1]. The Intergovernmental Panel on Climate Change in its Sixth Assessment Report illustrated that the global surface temperature will continue to rise until at least the mid-century, and extreme weather events such as hot extremes, heavy precipitation, and agricultural and ecological drought will be further intensified [2]. Changes in the global surface temperature and precipitation regimes have been affecting terrestrial ecosystems and species in many ways [3–5]. Such changes contribute to increased variability with a higher risk of rapid-onset climatic events, such as agricultural and ecological drought, storms, floods, wildfires, soil erosion, and forest dieback which would have a detrimental impact on forest ecosystems [6–8]. Even small changes in the climate attributes may result in vegetation shifts leading to significant changes in species composition [9–11], species distribution [12], productivity [13], and biomass production [11,14] including tree decline [15,16]. Projected climate change, combined with non-climatic drivers, may eventually cause the loss and degradation of much of the world's forests [1]. Despite the fact that trees and forests are major means of combating climate change, this is a major risk for forest health and vitality [17,18].

Recent studies and reports have indicated that Nepal is highly vulnerable to climate change. Warming in Nepal, especially in the High Mountains, is projected to be higher than the global average. Under the highest emission scenario, RCP8.5, the country is projected to warm by 1.2–4.2 °C by 2080 [19]. The long-term Climate Risk Index 2021 identified Nepal as one of the ten most affected countries by the impacts of extreme weather events over the last two decades [20]. The index focuses on extreme weather events but does not consider an important slow-onset process evident in Nepal, i.e., glacier melting. Extreme weather events such as drought, heatwaves, river flooding, and glacial lake outburst are all projected to intensify over the 21st century [19]. Compared to the rise in average temperature, rises in maximum and minimum temperatures are expected to be stronger in Nepal, and temperature increase is expected to be strongest during the winter months. Similarly, the changes in the distribution of precipitation among the regions have been stronger with both, negative and positive trends compared to the changes to the historical annual precipitation rate across the country. Bohlinger and Sorteberg [21], who analyzed the data from 98 meteorological stations across the country, found that the amount of precipitation and the number of extreme events per station are neither significantly increasing nor decreasing between 1971 and 2010. However, they noticed a systematic increase in the frequency and intensity of extreme precipitation events in the western part of the country. Additionally, Dahal et al. [22] using 32 years of monthly precipitation data (1981–2012) from 40 weather stations, suggested that there has been an increase in the intensity and frequency of drought, with the trend being stronger for longer droughts.

Zhang et al., [23] reported significant trends of increased frequency of hot days and concurrent decrease in soil moisture content over a large part of East Asia including eastern and central Nepal. They revealed a substantial shift toward a "drier–hotter" climate regime from the late 20th century onward in the region. Recognition of the occurrence of such a substantial shift in climate is crucial for the timely development of effective policy responses to cope with the change [24].

Some of the identified impacts of climate change on forests are (i) species' ranges are shifting to higher altitudes, for example, *Abies spectabilis*, and *Betula utilis*, (ii) increased incidence of forest fires in recent years, (iii) changes in phenological cycles of tree species, (iv) increased emergence and spread of invasive alien plant species, and (v) higher incidences of pests and diseases [25–29]. Many tree species move to a higher elevation in mountain regions as the climate warms [26]. However, tree species occurring near the mountaintops will have no space for their upward expansions in a warming climate. This makes high mountain forest ecosystems vulnerable to warming [30,31]. Global warming had resulted not only in the shifting of vegetation toward higher altitudes but also contributed to changes in phenology and regeneration of many dominant tree species. Even a small increase in the mean annual temperature, for example, of 1 °C brings substantial changes in the growth and regeneration capacity of many tree species [32]. There was less natural regeneration of *Abies spectabilis* due to radiation and low moisture [33].

Therefore, an enhanced understanding of how tree species dynamics change with a major shift in climate attributes is needed to develop adaptive management strategies to increase the resilience of forest ecosystems [34]. Dendroclimatology provides important information about past climate change and facilitates the understanding of the impacts of future climate on forest conditions [35]. Tree growth indirectly records an event that occurred in the environment in which it grows [36]. Tree rings are natural archives that preserve evidence of past climate changes. Tree-ring chronology data quantify the annual xylem growth of individual trees over their whole life span [37]. Temporal variability of temperature and precipitation regime results in site-specific and species-specific variations of tree-ring width [38], providing the opportunity to evaluate the growth patterns of the tree species under the changing climate [39]. Understanding the response of tree species is essential to comprehend their ability to adapt to changing climate, and therefore, increasingly important for forest management. There is growing evidence that tree species

diversity is closely tied to a wide array of ecosystem functions and services including resilience [18,34,40–42]. The Himalayas and high mountains constitute an important global biodiversity hotspot, yet studies on species' response to climate change from this region are lacking [12].

Hence, dendrochronological data presents the growth trends in a long historical perspective, i.e., a few hundred years, with an annual resolution. Tree rings as proxy-based information have been used for the reconstruction of temperature and precipitation variability, tree-ring-based drought reconstructions, and several other purposes since the early 20th century. Several studies elsewhere in the country have used tree ring-based reconstruction of climate indices, most often temperature, precipitation, and drought, to detect the impact of past climatic variability on tree growth [43–48]. However, only a few studies were conducted in the high altitude of the eastern Himalayan region of Nepal [48]. The overarching objective of this study was to explore how different tree species inhabiting together respond to the changes in climatic variables. The study included three tree species, i.e., *Abies pindrow*, *Pinus wallichiana* and *Tsuga dumosa*, which are considered climate-sensitive species in the temperate region of Nepal and investigated their growth performance in response to the changes in climate attributes. The study was conducted in one of the high-altitude mountainous districts of Nepal. The samples (tree cores) were collected from the trees located up to 3100 m high (1852–3106 m). Specifically, it examined the radial growth differences among the three-tree species, determines the relationship between tree growth and climatological conditions, and explores species-specific growth sensitivity to changing climate. This research will improve our understanding of the long-term effect of climate change on forest productivity.

## 2. Materials and Methods

### 2.1. Selected Tree Species and Study Area

For the study, three tree species *Abies pindrow* (Himalayan fir), *Pinus wallichiana* (Himalayan pine) and *Tsuga dumosa* (Himalayan hemlock) were selected. In Nepal, *Abies pindrow* mostly occurred in the western region from 2100 m to 3000 m, but the species is also found in the eastern–central highlands of the country including Dolakha District. It is usually dominant where it occurs, but also forms mixed stands with other conifers such as *Tsuga dumosa* and broadleaved trees, for example, *Juglans* and *Quercus* [49]. *Pinus wallichiana* is found in Nepal between 1800 and 3600 m, and very occasionally to 4400 m. The species is widely distributed in the midland zone between the foothills and the main Himalayan range, where it is often mixed with *Pinus roxburghii*. *Tsuga dumosa* is a species of conifer native to the eastern Himalayas and is found between 2100 and 3600 m altitude in Nepal. In higher altitudes, this is often found as an associate of *P. wallichiana* [49].

Dolakha District, located in a mountain physiographic area in the central east part of Nepal (Figure 1), was selected for the study. The district is ranked highly vulnerable in Combined Risk Index including flood, drought, landslide, ecological, temperature, rainfall and glacier lake outburst flood [50,51]. For the last 12 years, the monthly average day temperature ranges from 3 °C in January to 16 °C in June. Likewise, the monthly average night temperature ranges from −7 °C in January to 9 °C in July. The average monthly precipitation for the same period ranges from 6.3 mm in November to 352.3 mm in July [52].

The samples for this study were collected from five community forests (CFs) located in the district. Community forestry is one of the six community-based forest management models practiced in Nepal. In this model, local people are organized in a group, commonly known as a community forest user group (CFUG), and take the responsibility of managing forest resources [53]. The CFUG prepares and executes the periodic forest management plan and the CFUG constitution in order to manage their forest sustainably. The five CFs selected for this study were: Hanumante CF, Bhatekhola CF, Kalobhir CF, Jireshwori CF and Debithan Kimane CF (Table 1). The forests are natural forests and are managed by the respective CFUGs. A rigorous discussion was held with Division Forest Office (DFO) personnel in order to select the CFs. The CFs were selected based on the criteria:

(a) accessibility-easy to access rural, remote mountainous CFs, (b) permission for sample collection, and (c) scale of the fieldwork support from the DFO and local forest officers. The study area is located at $27°6'$ N latitudes and $86°2'$ E longitudes. The forest type is a temperate and sub-alpine mixed forest dominated by *Abies pindrow*, *Pinus wallichiana* and *Tsuga dumosa*.

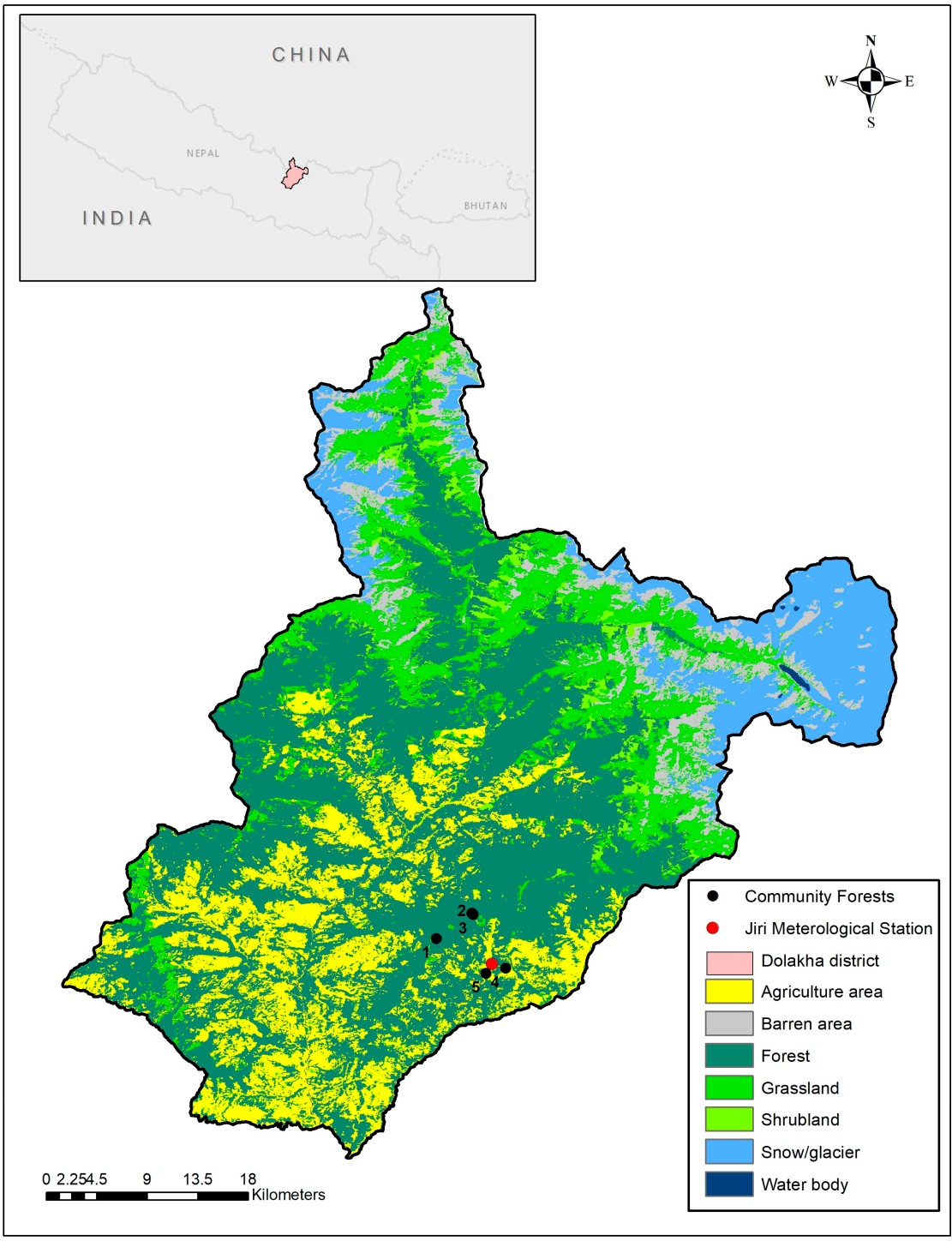

**Figure 1.** Map showing the location and outline of the study area Jiri, Nepal. The red-filled circle shows the location of the Jiri meteorological station (27.6304 E, 86.2321 N; 2003 m). Black filled circles indicate the location of the five community forests (CFs) from where samples were collected, (1) Hanumante Community Forest (CF), (2) Bhatekhola CF, (3) Kalobhir CF, (4) Jireshwori CF, and (5) Debithan Kimane CF.

**Table 1.** Location and elevation (m) of community forests from where samples were collected.

| Forest Name | Elevation (m) | Latitude (N) | Longitude (E) |
|---|---|---|---|
| Hanumante Community Forest | 2833 | 27.6508 | 86.1873 |
| Bhatekhola Community Forest | 2742 | 27.6712 | 86.2160 |
| Kalobhir Community Forest | 2769 | 27.6703 | 86.2169 |
| Jireshwori Community Forest | 1953 | 27.6269 | 86.2429 |
| Debithan Kimane Community Forest | 1903 | 27.6227 | 86.2271 |

### 2.2. Sample Collection and Preparation

Samples for the study were collected from 30 *Abies pindrow*, 27 *Pinus wallichiana*, and 27 *Tsuga dumosa* trees. The elevation of the sample tree location ranges from 1852 m to 3016 m. Table S1 (Supplementary Material) provides information on the location (forest name, altitude, slope, aspect, latitude and longitude) of all sample trees. Samples were collected randomly from dominant trees from each species. To minimize the effect of other micro-environmental factors (such as competition for light, moisture, and nutrient) on tree species, dominant trees were selected. A five mm increment borer was used to collect two cores of each tree. Increment cores were extracted from breast height (1.3 m above ground) in two directions with an angle of 90°. Information on site attributes such as altitude, slope gradient, aspect, and tree attributes such as the location of each tree, species, diameter at breast height, total tree height, and crown-base height was recorded during the sampling. The collected samples were labelled (Tree code, Tree ID, Core codes) and transported to the dendrology laboratory. The cores were left for air-drying before further analysis.

### 2.3. Tree Ring-Width Measurement and Analysis

After air drying, the tree cores were cleaned with razor blades to improve the visibility of tree ring boundaries. Individual tree ring was marked, and ring width was measured according to the standard dendrochronological techniques. Tree ring width was measured with a LINTAB (Linear positioning table) measuring system, and a binocular microscope connected to a PC with an accuracy of 0.01 mm. Ring widths were recorded using the TSAP-win (time series analysis package for windows) software package [54].

Two cores of the same tree were cross-dated to detect apparent measurement errors such as missing rings. Afterwards, COFECHA [55] was used to check the accuracy of measurement and cross-date. COEFECHA was used iteratively as needed to re-examine the flagged increment cores.

### 2.4. Climate Data

Climate data of Jiri meteorological station (27.6° N, 86.2° E, 2003 m elevation) were obtained from the Department of Hydrology and Meteorology, Kathmandu, Nepal. Data were available for different climate variables (e.g., monthly maximum air temperature, monthly minimum air temperature, monthly mean air temperature, monthly precipitation). The data were acquired for the period of 1966 to 2021. For the study, to compare species-specific responses to the climatic variable, we chose the same time window, i.e., 1973 to 2021. There was a few months with missing values (3%–4%) for both temperature and precipitation. To compute the missing data, we downloaded ERA5 data from Copernicus Climate Change Service (C3S 2021). ERA5 data were then used to reproduce observed (measured) values. A statistical correction was performed to fix the systematic difference between reanalysis and observation values.

### 2.5. Standardization and Chronology Building

All measured tree ring series were standardized by applying a double detrending process using the dplR package [56] based on R software [57]. Firstly, modified negative exponential curves were applied to remove geometric growth trends. Then smoothing cubic spline with a 50% wavelength cutoff equal to 67% of the series length was fitted to

remove the age-related growth trends and to preserve a common signal for climate growth analysis [58]. The standardized site chronology still contains autocorrelation because of the impact of the previous year's climate on the subsequent year's tree growth. We use autoregressive models with a maximum order of three to remove autocorrelation of every individual series and to ensure the obtained chronology contains only tree ring variability related to climate variability. All detrended series were then averaged (arithmetic mean) to develop the ring-width chronology for that site. The dplR was further used to compute various descriptive statistics such as Expressed Population Signal (EPS), mean index, standard deviation, mean sensitivity, series inter-correlation, autocorrelation, and mean inter-series correlation (Rbar). Expressed Population Signal (EPS) was used to test the population strengths in the resulting tree-ring chronologies [35]. Mean sensitivity (MS) is a measure of the year-to-year variability in tree ring width and was employed as a measure for variability in the tree-ring series [35]. Series inter-correlation indicates the site/stand-level signal. Auto-correlation (AC) was calculated to assess the influence of the previous year's growth on the current year's growth [59]. The mean inter-series correlation (Rbar) was calculated to assess the signal strength throughout the chronology [60].

*2.6. Climate Growth Relationship*

We analyzed the annual trend of temperature and precipitation using the Mann—Kendall test [61,62] and Sen's slope [63]. Bootstrapped correlation function was applied to assess the relationship between tree-ring chronologies of each species and climate variables (monthly mean temperature, and monthly precipitation sums). Correlation analysis was performed using a 19-month window starting from May previous year till November current year [64]. Part of the previous year was included in the analysis because of the lag effect that climatic parameters can have on tree growth in the following year [65]. The stationarity and consistency of the climate growth relationship over time were explored using moving correlations analysis. The analysis was completed using the R package treeclim [56,66].

## 3. Results

*3.1. Chronology Characteristics*

Out of 30 collected samples, 24 *A. pindrow* trees (45 cores) were successfully cross-dated. The chronology spanned 94 years (1928–2021). For *P. wallichiana*, out of 27 samples collected, we were able to cross-date only 23 trees (41 Cores). The chronology spanned 49 years (1973–2021). For *T. dumosa*, out of 27 sampled trees, 24 trees (44 Cores) were successfully cross-dated. The chronology spanned 61 years (1961–2021). The residual chronology of each species is presented in Figure 2.

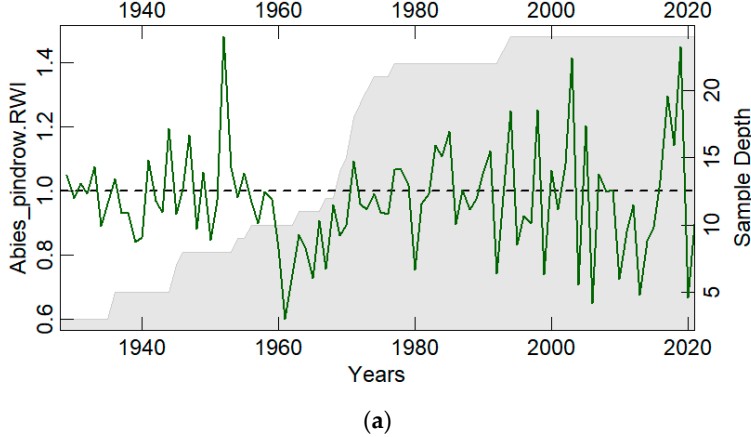

(**a**)

**Figure 2.** *Cont.*

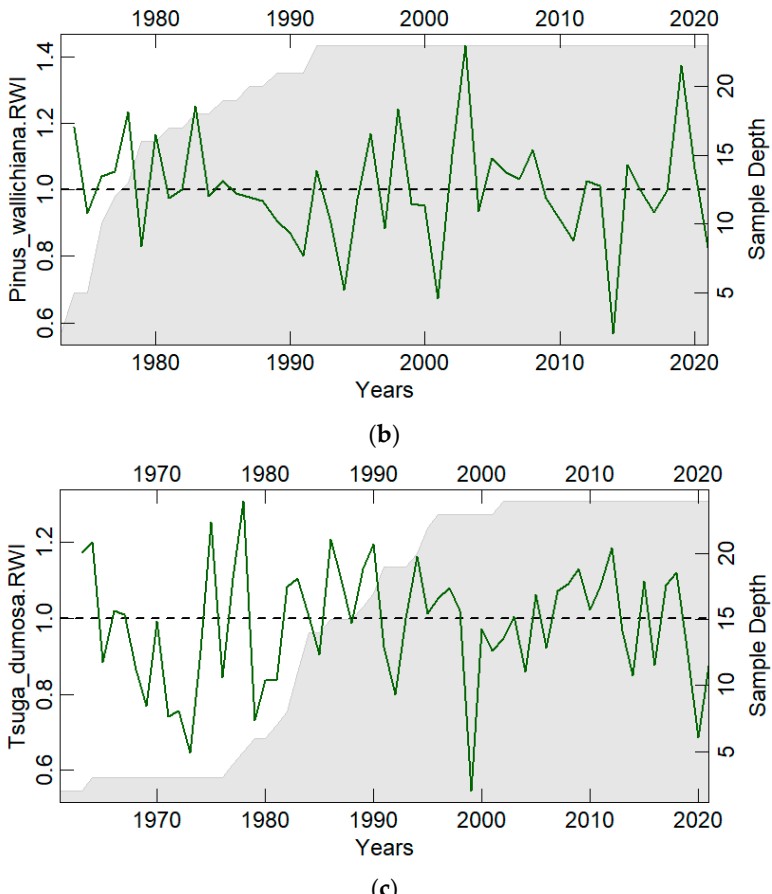

**Figure 2.** Residual chronology of *Abies pindrow* (**a**), *Pinus wallichiana* (**b**) and *Tsuga dumosa* (**c**) from Jiri, Nepal. Grey shaded area indicates sample depth.

Dendrological parameters for each of the studied species are shown in Table 2. The mean sensitivity of the species ranged from 0.24 (*T. dumosa*) to 0.34 (*P. wallichiana*). Series intercorrelation, which represents the common stand-level signal recorded of a site ranged from 0.472 (*A. pindrow*) to 0.586 (*T. dumosa*). Expressed population signal ranged from 0.853 (*A. pindrow*) to 0.888 (*P. wallichiana*), which is above the threshold value of 0.85. Mean sensitivity, mean series intercorrelation, and EPS indicated the strength of chronology for climate growth analysis. The mean tree ring width of *T. dumosa* showed the maximum growth rate (5.31 mm) among the species, while *A. pindrow* showed the lowest mean tree ring width per year (3.50 mm). The higher ring-width variability was observed in *P. wallichiana* followed by *T. dumosa*. Autocorrelation observed was low for all three species. Both autocorrelation and Rbar are in a reasonable range compared to similar tree-ring studies.

**Table 2.** Dendrological parameters of *Abies pindrow*, *Pinus wallichiana*, and *Tsuga dumosa* in Jiri, Nepal.

| Parameters | Abies pindrow | Pinus wallichiana | Tsuga dumosa |
|---|---|---|---|
| No. of trees/cores | 24/45 | 23/41 | 24/44 |
| Overall period | 1928–2021 | 1973–2021 | 1961–2021 |
| Age range (years) | 94 | 49 | 61 |
| Mean tree ring width (mm) | 3.50 | 4.578 | 5.31 |
| Standard deviation (mm) | 1.80 | 2.96 | 2.10 |
| Mean sensitivity | 0.30 | 0.34 | 0.24 |
| Series intercorrelation | 0.472 | 0.517 | 0.586 |
| Autocorrelation | 0.24 | 0.175 | 0.247 |
| Expressed Population Signal (EPS) | 0.853 | 0.888 | 0.868 |
| Rbar | 0.268 | 0.284 | 0.304 |
| Mean GLK * | 0.624 | 0.98 | 0.928 |

* GLK: Gleichläufigkeit (sign test).

### 3.2. Local Climate

As recorded from the meteorological station, the mean monthly temperature of Jiri meteorological station was 14.3 °C and the mean monthly precipitation sum was 196.54 mm during the analyzed period (1973 to 2021) (Figure 3). June was the warmest month and December was the coldest month. July received the highest precipitation in the area and December was the month with the lowest precipitation. The annual trends of both temperature and precipitation show there is an increasing trend slightly after 1980. Mann–Kendall test also showed a significant increasing trend for both temperature (Z = 3.5661, $p < 0.001$) and precipitation (Z = 3.596, $p < 0.001$).

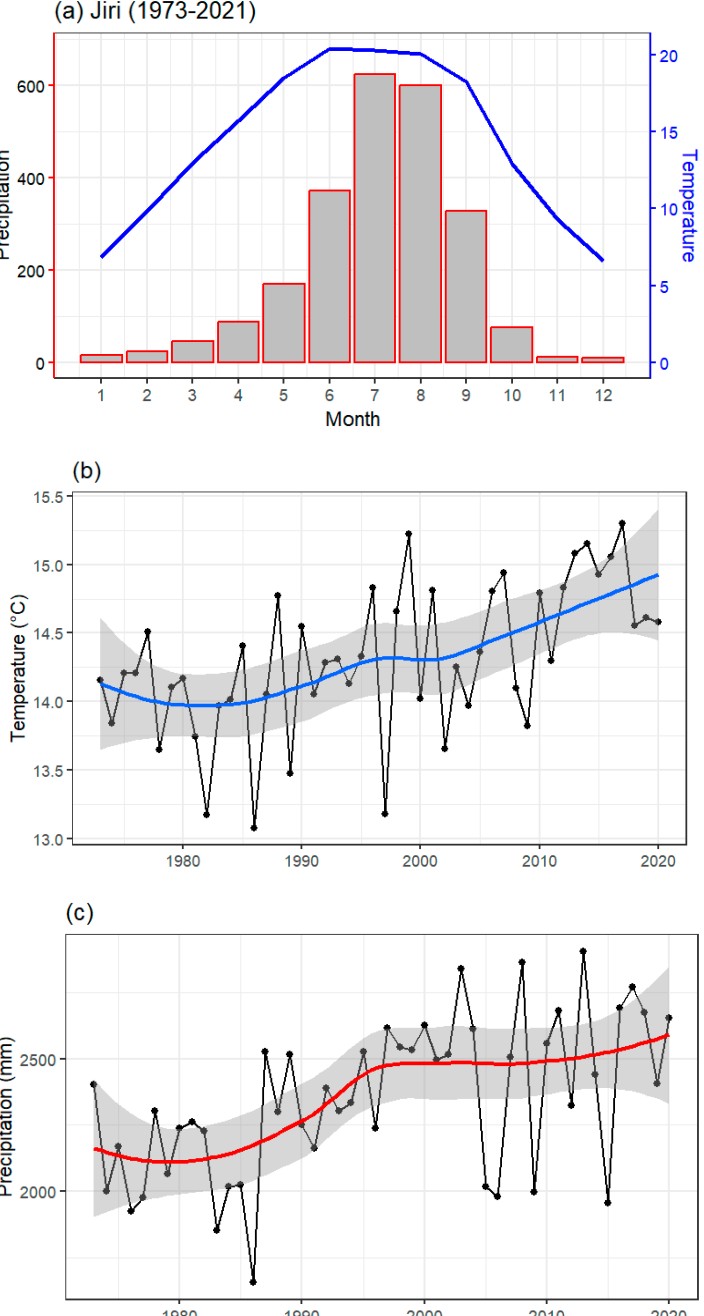

**Figure 3.** Trends of climate attributes for Jiri Nepal. (**a**) Climate diagram for Jiri, Nepal shows the mean monthly temperature (line) and the total monthly precipitation (bar). Months are numerically represented with January being 1 and December being 12. (**b**) Annual mean temperature. (**c**) Annual precipitation sum.

### 3.3. Tree Growth Response to Climatic Variable

The bootstrapped correlation function between the tree ring chronology and climate showed that the growth of *A. pindrow* was mainly stimulated by the high temperature in June (Figure 4). Though there was no significant correlation between summer temperature and the growth of *P. wallichiana*, the bootstrapped correlation function analysis suggests the negative correlation of the tree ring growth with previous years' December temperature. A negative correlation is observed between the tree ring growth of *T. dumosa* and March temperature.

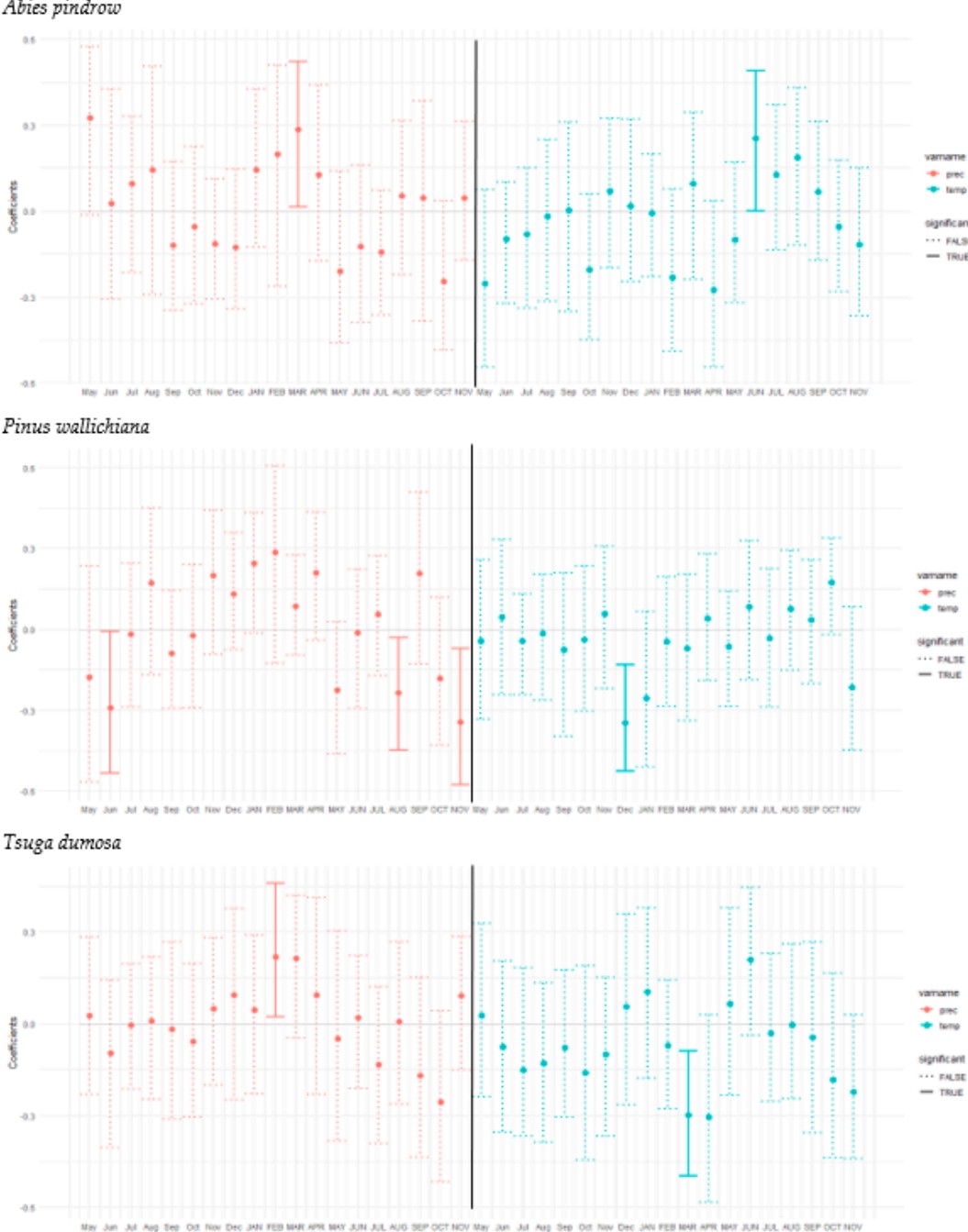

**Figure 4.** Bootstrapped correlation function analysis of tree ring growth of *Abies pindrow*, *Pinus wallichiana*, and *Tsuga dumosa* to sums of precipitation and mean monthly temperature from previous May to current November for Jiri, Nepal. Months from the previous years are denoted with lowercase letters, while months in the current year are denoted with uppercase letters.

Tree ring growth of the *A. pindrow* had a significant positive correlation with March precipitation. Tree ring growth of the *P. wallichiana* was stimulated by the lower precipitation in June of the previous year and August and November of the current year. Tree ring growth of the *T. dumosa* was positively correlated with February precipitation (Figure 4).

The stationarity and consistency of the climate growth relationship over time were explored using moving correlations analysis (Figure 5). In the case of *Abies pindrow*, moving correlation confirmed that June temperature has the stationery response over time. April temperature and December temperature have also produced significant negative transient responses on the growth of the *A. pindrow*, which diminished over time. In recent years, a strong correlation between tree ring growth and higher August temperature has been observed. With precipitation, the moving correlation analysis of the climate growth relationship of *A. pindrow* showed a consistent and stationary relationship over time. March precipitation showed a significant positive impact on tree growth since 1986.

*Abies pindrow*

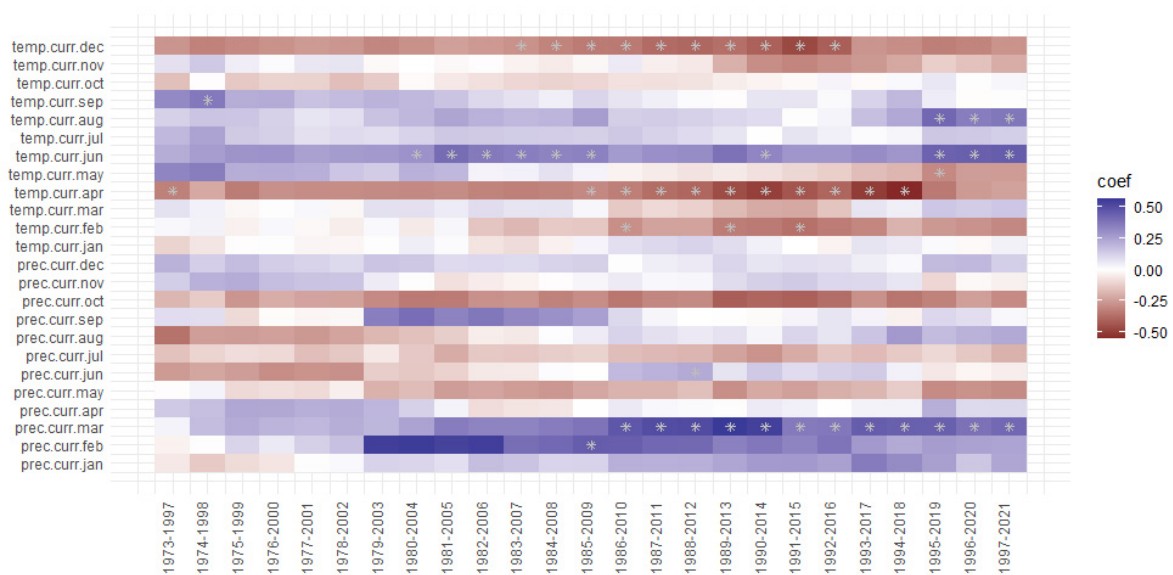

*Pinus wallichiana*

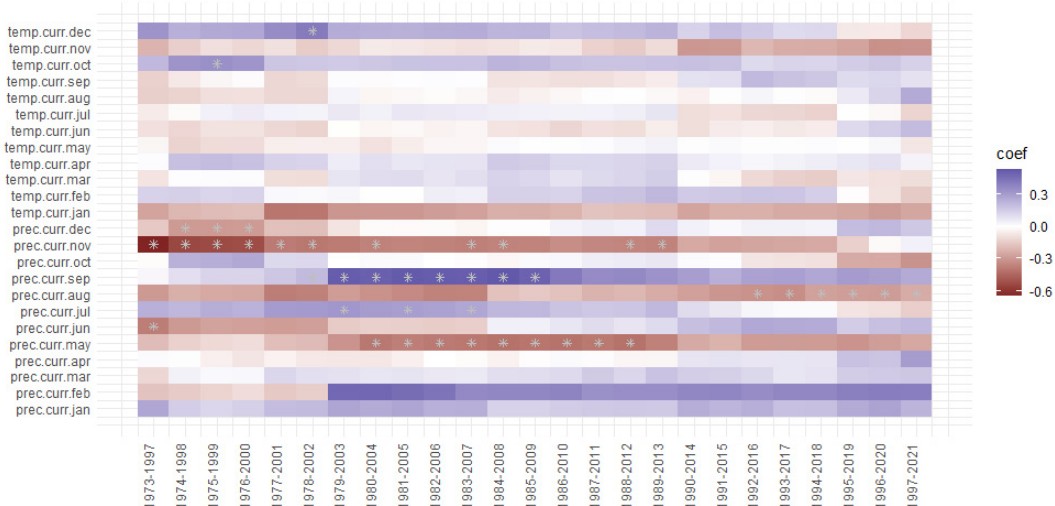

**Figure 5.** *Cont.*

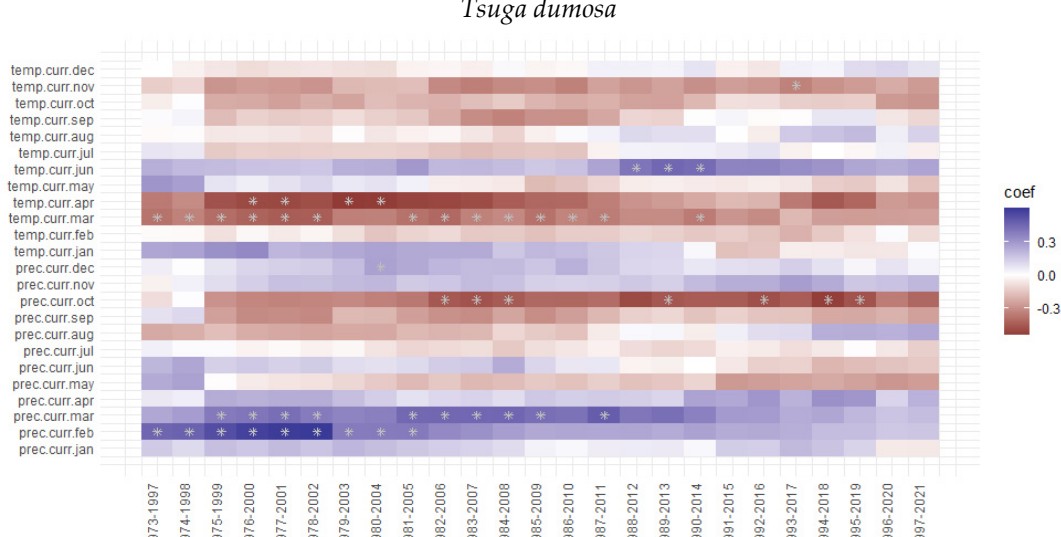

**Figure 5.** Moving correlations (25-year window) between tree ring growth of *Abies pindrow*, *Pinus wallichiana*, and *Tsuga dumosa* and climatic variables (mean temperature and precipitation) for the period of 1973 to 2021. The color coding signifies the correlation coefficient, and significant correlations are symbolized by white asterisks.

In *Pinus wallichiana*, the moving correlation analysis showed that tree ring growth is not influenced by temperature in recent times (Figure 5). However, with precipitation, the relationship has been inconsistent. May precipitation showed a significant negative impact on tree growth through the period 1976 to 2013: however, the effect diminished in recent years. In recent years, a stronger negative correlation is being seen between tree growth and August temperature with the decreased impact of May precipitation.

In *Tsuga dumosa*, the moving correlation analysis showed again the relationship of tree growth with both temperature and precipitation is inconsistent over time (Figure 5). During the period from 1973 to 2011, winter precipitation (February and March) had a positive impact on tree growth: however, the trend declined over the years. For years, October precipitation had a significant negative impact on the growth of *T. dumosa*, again the effect disappeared in recent years.

## 4. Discussion

Most of the chronology statistics obtained for *A. pindrow* in the current study are similar to the previous study in the species [67]. Similarly, chronology statistics for *P. wallichiana* and *T. dumosa* are of similar magnitude as those reported for the species [47,68] and other temperate species [69].

The chronology signals such as mean sensitivity, EPS, and series intercorrelation for all three species from Jiri suggested that they have valuable climatic signals for the reconstruction of past climate. The statistical data confirm the relevance of using these data to study tree ring-climate relationships. The growth of *Abies pindrow* was mainly stimulated by the high temperature in June indicating that the species thrive better in warmer summer temperatures. Studies have shown that the growth of the higher altitude species is favored by higher summer temperatures [70]. The growth of higher altitude trees may be limited by temperature, so an increase in temperature may increase tree growth in many species up to a point. Higher temperatures may increase the rate of photosynthesis and hence may enhance the tree ring growth [71]. Most of the wood formation process in *A. pindrow* occurs from April to September. This clearly shows the importance of summer temperature for radial growth in temperate regions [67]. Furthermore, the increase in summer temperature may result in a longer growing season, which again may result in increased radial growth of higher altitude species [72]. In the studies conducted in high-altitude temperate regions,

it has been observed that increased temperature during early spring can stimulate the cambial activity and prolong the growing season resulting in wider tree ring widths [73,74]. However, in our study radial growth of *P. wallichiana* and *T. dumosa* was not affected by the summer temperature. Growth of *P. wallichiana* responded negatively to the previous year's December temperature. In a temperate region, trees are dormant during winter. Higher temperature in winter may result in high respiratory loss of food reserves so is detrimental to growth of temperate species [75]. In the same study, Yadav and Bhattacharyya [75] observed the opposite, winter temperature of the previous year had positive impact on growth of *P. wallichiana*. Looking back to climate scenario of the study area, the mean temperature has been increased by nearly 1.5 °C in just 40 years interval.

Growth of *Tsuga dumosa* responded negatively to the current year's March temperature. Such a negative correlation between radial growth and spring temperature in *T. dumosa* has been reported in previous studies [47,76]. Studies have reported that the increase in pre-monsoon temperature has had an adverse impact on the growth of most of the conifers in the temperate region [77–79]. Many studies have reported the additive effect of temperature and precipitation during the winter and spring season on the growth of *T. dumosa* [47], i.e., the growth of *T. dumosa* have a positive correlation with spring precipitation and a negative correlation with winter and early spring temperature [76,80]. Our study also has a similar result. We observed a rise in February precipitation stimulates the growth of *T. dumosa* and a rise in March precipitation stimulates the growth of *A. pindrow*. In the study conducted in *Abies pindrow* from western Nepal (36) and Jammu and Kashmir [81], a significant positive relationship was observed between tree ring width growth and precipitation in March, April, and May. As winter and early spring months are mainly dry in this region, precipitation may increase soil moisture available at the root zone for the early growing season and hence may favor the growth of the species [79]. In contrast, an increase in temperature in the early growing season may increase soil moisture evaporation and plant respiration, which may lead to the reduction of carbon absorption for radial tree growth [80].

The dynamic of correlation between the precipitation and the growth of *P. wallichiana* is again different to the other two studied species. Tree ring growth of *P. wallichiana* responded negatively to the current year's August and November precipitation. However, the impact of November precipitation is declining, and strong signals are developing for August precipitation. Similarly, a negative correlation between the previous year's June precipitation and growth is observed. In one of the earliest studies on the climate growth relationship of *P. wallichiana*, Yadav and Bhattacharyya [75] also reported November precipitation has a negative effect on the growth of the species. In the current scenario of increasing precipitation and temperature trend of the site, the study suggests that climate change could have an adverse impact on the growth of *P. wallichiana*.

Understanding the resilience of the forests of *P. wallichiana*, *A. pindrow* and *T. dumosa* is important for managing and understanding the mountain ecosystems and local communities. The present study provides strong evidence of the impact of climate change, especially temperature and precipitation, on the growth of the three-tree species. Evidence suggested that the increasing amount of soil organic carbon storage as altitude increases in the forests of Nepal [82]. This implies that high-altitude forests are critical repositories of carbon and indicated the importance of conservation and wise use of high-altitude forests in the context of forest-based solutions to mitigate climate change. As the rate of the climate change (warming) in the Himalayas and the High Mountains has been found to increase, the forest ecosystem biomass and carbon across the high mountain regions may respond differently than the forest ecosystems of other geographic regions. The resilience of the mountain forest ecosystems which are exposed to increased warming and changes in rainfall patterns and closed to higher human interventions need to be explored. There is, hence, a strong need for increased investment in research to narrow the knowledge gaps- how tree species are responding to climate change, and forests are recovering under different recovery contexts. The Government of Nepal needs to intervene with silviculture treatments and

forest management plans in close coordination with the community forestry user groups. Silvicultural intervention such as thinning increases forest resiliency during drought and drought-induced water stress [83,84]. Selective logging of over-matured trees within the limit of the annual allowable cut prescribed in the CFUG forest management plan might bring multiple benefits, for example, revenue from forest management while promoting natural regeneration including recruitment. The species-specific response to climate change, for example, growth and reproduction are important vegetation response matrices to be considered while designing forest management plans and silvicultural treatments.

## 5. Conclusions

The study emphasized the importance of the Himalayan conifers in recording the variability in climatic variables. Furthermore, it also provides evidence that the tree growth relationship with climatic variables is species specific. While the climate and growth of *Pinus wallichiana* and *Tsuga dumosa* revealed some annual and decadal trends but did not show any strong long-term consistent trend. However, *Abies pindrow* showed a strong positive relationship between March precipitation and June temperature. This positive relationship is consistent and stationary over time. Studies on *Abies pindrow,* if extended to other sites, would be interesting to see how the species respond to climate change in different environments. In the context of increasing warming, increasing trends in the frequency and intensity of drought, and highly biodiverse forests in the high mountain ecosystems, a great deal of empirical and experimental research work is needed to evaluate forest resilience under current and projected environmental changes and to explore the post-disturbance forest management practices to enhance the forest resilience.

**Supplementary Materials:** The following are available online at https://www.mdpi.com/article/10.3390/f14040737/s1, Table S1: Location of the sample trees (Forest name, Altitude, Aspect, Latitude and Longitude).

**Author Contributions:** P.R.N., A.G. and M.K. conceived, designed, and collected the samples for the study. P.R.N. and A.G. analyzed the data and wrote the manuscript. M.K. acquired the funding, approvals, supervised, reviewed, and edited the manuscript. R.K. and B.S.P. managed fieldwork and reviewed and edited the manuscript. All authors have read and agreed to the published version of the manuscript.

**Funding:** The project was partially funded by the Deutsche Forschungsgemeinschaft (DFG, German Research Foundation) under Germany's Excellence Strategy—EXC 2037 'CLICCS—Climate, Climatic Change, and Society'—Project Number: 390683824.

**Data Availability Statement:** All data relevant to the study are included in the article.

**Acknowledgments:** We would like to thank Anna Klemmer for her technical assistance in the lab. We would like to thank Benjamin Poschlod, Kumar Darjee and Leam Mykel Martes for helping with missing data interpolation using ERA5 data. We are grateful to Krishna Prasad Acharya, Shiva Sapkota, Manoj Ghimire, Suman Sapkota, Bhuwan Timilsina, Rojan Pokharel and other staff members of Division Forest Office, Dolakha for their help in collecting and handling samples.

**Conflicts of Interest:** The authors declare no conflict of Interest.

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
