# Peer review of "Species-Specific Response to Climate Change: Evident through Retrospective Analysis Using Tree Ring Data"

_forests, doi:10.3390/f14040737_

Round 1
Reviewer 1 Report
General comments
The paper entitled "Species-Specific Response to Climate Change: Evident Through Retrospective Analysis Using Tree Ring Data" proposes an original study to characterize the relationships between three tree species distributed at very high altitudes in the Nepalese highlands and climate. The objective of this study is to investigate the response of trees to climate and their ability to adapt to climate change using tree-ring data. This paper proposes original results that deserve to be published after major revisions have been made.
I think the introduction and discussion need serious revision and improvement. Please find all of my recommendations about these aspects below. I also suggest some specific comments.
Abstract:
L.14-16: You do not address either disruption or resilience in this paper it seems strange to me that these specific terms would introduce the abstract, I would expect you to explore these aspects in the paper after reading this abstract.
L. 23-25 : Why not just use response functions or correlations?
L. 27-29 : There is a missing aperture at the end of this summary.
Introduction
Introduction construction problem. I suggest you reorganize your paragraphs and start with the climate drift from the global to Nepal (paragraph 1 and 4) then talk about the interest of tree rings (paragraph 3 and 2), then finally paragraphs 5 and 6 (so in order 1, 4, 3, 2, 5, 6). It is also important that you bring more clearly the assumptions and objectives of this study that are still obscure in your introduction
L. 72: With the merits => I don’t understand what do you mean? Is it necessary?
L. 100-101: What types of impacts on forest ecosystems did changes in climate attributes already involve? Can you give some references?
L. 103: draught => Are you sure of the term?
L. 108-119: This last paragraph needs to be improved in order to better introduce assumptions and questions to which the study aim to answer.
Methods and material
The methods are well detailed, but the study area remain poorly described.
L. 123-128: I would have highly appreciated a more detailed description of the study area including the location of trees (are the three species distributed along a gradient, or are they located randomly?), slopes, geology, soil profile, etc….
Moreover, I do not know what a community forest in Nepal is and how does it manage by people.
L. 138-142: I suggest removing this section to the previous paragraph and may be add a table with a synthesis of all attributes.
L. 184-187: I would appreciate you explain why you compute such statistics, to what objective?
L. 191: I don’t understand why you do not have use response function also to assess climate-growth relationships. It is quite strange for me you used bootstrapped correlation and then moving correlation. I suggest using fix and mobile either response function or correlation.
Results
The results are fairly well presented and clear, but the figures deserve some revision.
L. 232-237: I am really surprised about the low amount of precipitation in winter. I wonder about snow cover which may play a key role for tree growth.
L. 250-260: I think it would be easier to read if you present the results for each species successively.
Fig 1 et 2: I suggest increasing police size in axis labels and tick labels. It would be interesting to compare the evolution of temperature and precipitation to the RWI evolution of the three species
Fig 3: Graphics need some improvement. Change the legend on the left. I suggest also increasing police size and adding species name in the bottom of each graphics. Moreover, it seems to me more intuitive to represent precipitation in blue, and temperature in red.
Fig 4: As for Fig. 3, please add the species name in the bottom of each graphics. Moreover, it will be easier to read the graph if you separate precipitation and temperature by a black line. The moving response function with previous year are not presented here, why?
Discussion
Overall, the discussion needs to be improved. I'm still waiting to hear why some species respond differently than what has already been observed elsewhere? What are the specific attributes of the study area? Why do two of the three species not respond to summer temperature? Are there ecophysiological reasons?
There is no need for two sections in this paragraph that are unequal by the way.
It seems to me that this discussion lacks criticism and hindsight (rather short tree-ring chronology, only one forest studied, only a few trees). Moreover, the strict contribution of this study is difficult to perceive. It would be interesting to compare your results with other large massifs such as the Alps to show the similarities/differences.
L; 340-341: You mention an increasing precipitation, so why Abies exhibit higher response to precipitation in mars? Is such results already reported for the species?
L. 348-349: Where do you exhibit a decrease in August and November precipitation? If I am not wrong, you do not have presented a detailed description of the climate evolution in term of monthly changes in mean temperature or precipitation?
L; 361-363: I think that such conclusions is difficult to demonstrate based only on the few data you presented. To confirm this assumption, you need to replace all studies done on these species in different context and demonstrate that whatever the conditions trees of the three species became more and more sensitive to climate.
L. 363-364: What is the interest of this sentence?
L. 375-381: Beware, many recent articles suggest that close-to-nature management is better suited to promote carbon sequestration and mitigate climate change. There is still a nice debate about which treatments and silvicultural practices are best. Your study does not address this issue, but you can discuss these aspects in light of your results => Abies is increasingly sensitive to drought => after heavy harvests, the forest environment is drier increasing the risk of drought-induced mortality.
Conclusion:
I think you should add an opening question at the end of the conclusion, somethings about the resilience of forest to extreme events to complete your analysis of climate growth relationships.
Reviewer 2 Report
Dear Editor.
I have finished my review on the proposed paper “Species-Specific Response to Climate Change: Evident Through Retrospective Analysis Using Tree Ring Data”, forests-2222114-peer-review-v1.
Summary of the manuscript:
In the proposed paper, the author’s goal is to analyze how different species inhabiting together respond to changes in climatic variables using dendroclimatic analysis. They assessed the growth performance of Abies pindrow, Pinus wallichiana, and Tsuga dumosa in the temperate region of Nepal.
General review:
1. Generally, the manuscript presents an interesting topic and the specific research seems to include some significant points for the research community of this field.
2. The proposed paper is very well written with very good use of English language. Except some very minor grammatical mistakes and word errors. The author should check again the paper to correct these minor mistakes.
3. The proposed paper is very well structured. It begins with the Introduction with some references that helps the reader to get into the subject immediately. In Introduction there is an effort to provide previous studies with similar scientific content, which took place in the research area and in other countries. Author describes and set very well the scientific problem and how other researchers have approached. At the end of Introduction, authors clearly state the goals of the research. However, I believe that for the specific subject you can enhance the provided literature.
4. The methodology is generally very interesting. However, need some clarifications (see below comments).
5. The results and the discussion are generally OK. However, there some parts that need revisions (see below comments).
Additional points for revision:
In my opinion, the proposed paper could be characterized as a very good research work, complies with aims of FORESTS.
Lines 122-129, 2.1. Selected Tree Species and Study Area: Here, you provide some information about the study area. But most readers are not familiar with the study area. So, you should add a map with the study area and the exact locations of sampling, elevation, the expansion of the 3 studied species and some cities, in order to understand where is the study area.
Lines 160-171, 2.4. Climate Data: In the above map you should add the location of the meteorological station.
Lines 176-178: Here, you say that you retained the 50% of the variance. However, how many are the years of the wavelength that you used in the analysis.
Lines 184-187: How many years is the moving window and hoe many are the years of the overlap?
2.5. Standardization and Chronology Building: Generally, this section is weak. It needs more details and literature.
Line 212: “….0.85”. This value needs support with literature.
Lines 212-217: This part needs more scientific explanation and interpretation of the statistics.
Table 1: The Autocorrelation and the RBAR values are low. How you explain this?
Lines 236-237: The increasing trend of temperature and precipitation is based only on the moving average (figure 2b,c). This is a speculation. To prove if there is a trend you should apply a trend test, like Mann-Kendal.
Lines 322-322: Here, you should add a new phrase about the prolonged growing season. In Abies species, it has been proved that increased temperatures in temperate regions with high altitudes, during the early spring, can stimulate the cambium activity and prolonged the growing season, giving wider tree ring widths (doi.org/10.3390/f13060879 and doi.org/10.1016/j.dendro.2014.05.003). Please, add a phrase about this and use the proposed literature.
Round 2
Reviewer 1 Report
General comments
The paper entitled "Species-Specific Response to Climate Change: Evident Through Retrospective Analysis Using Tree Ring Data” has been greatly improved and I thank the authors for the work done which greatly improves the quality and ease of understanding of this very interesting paper, which deserves to be published in the journal Forest.
I recommend, nevertheless, some more corrections before publication.
Abstract:
I have no more comments on the summary which I find very good as it is.
Introduction
I still find that the passage that talks about Zhang at the beginning of the 3rd paragraph (L. 78-82) would deserve to join the 2nd paragraph (L. 73, juste before “Temperature variability […]. I would see a third paragraph already oriented on trees, beginning with the sentence L. 73-76.
I suggest you (by adapting a little the sentences and their sequence):
Zhang et al., [22] reported the significant trends of increased frequency of hot days and concurrent decrease in soil moisture content over a large part of East Asia including eastern and central Nepal. They revealed a substantial shift toward a “drier-hotter” climate regime from the late 20th century onward in the region.Recognition of the occurrence of such a substantial shift in climate is crucial for the timely development of effective policy responses to cope with the change [28].
Some of the identified impacts of climate change on forests and biodiversity are (i) species’ ranges are shifting to higher altitudes, for example, Abies spectabilis, and Betula utilis, (ii) increased incidence of forest fires in recent years, (iii) changes in phenological cycles of tree species, (iv) increased emergence and spread of invasive alien plant species, and (iv) higher incidences of pests and diseases [32]. Many tree species move to higher elevation in mountains regions as the climate warms [33]. However, tree species occurring near the mountain tops will have no space for their upward expansions in a warming climate. This makes high mountain forest ecosystems vulnerable to warming [34,35]. Global warming had resulted not only in shifting of vegetation toward higher altitude, but also contributed changes in phenology and regeneration of many dominant tree species. Even a small increase in the mean annual temperature, for example, of 1 °C brings substantial changes in the growth and regeneration capacity of many tree species [36]. There was less natural regeneration of Abies spectabilis due to radiation and low moisture [37].
Therefore, an enhanced understanding of how tree species dynamics change with a major shift in climate attributes is needed to develop adaptive management strategies to increase the resilience of forest ecosystems [23]. Dendroclimatology provides important information about past climate change and facilitates the understanding of the impacts of future climate on forest condition [29]. Tree growth indirectly records an event that occurred in the environment in which it grows [30]. Tree rings are natural archives that preserve evidence of past climate changes. Tree-ring chronology data quantify the annual xylem growth of individual trees over their whole life span [31]. Temporal variability of temperature and precipitation regime results in site-specific and species-specific variations of tree-ring width [20], providing the opportunity to evaluate the growth patterns of the tree species under the changing climate [21]. Understanding the resilience and resistance of tree species is essential to 85 comprehend their ability to adapt to changing climate, and therefore, increasingly important for forest management. There is growing evidence that tree species diversity is closely tied to a wide array of ecosystem functions and services including resilience [23- 27]. The Himalayas and high mountains constitute an important global biodiversity hotspot, yet studies on species’ response to climate change from this region are lacking [10].
Hence, dendrochronological data presents the growth trends in a long historical […]
L. 85: Do not use the terms resilience and resistance but "response".
L. 100-103: I don't understand why you are talking about climate reconstruction, droughts, that's not what you are studying
L. 128-131: I find the first sentence totally useless here and redundant with all that has been said previously. While the second one is totally out of subject and context....
L. 131: remove “biodiversity”
L. 132-133: add at least one reference…
L. 131-144: As you could see in the proposal 'sequence', I suggest you to go back to this paragraph much earlier in the introduction
Methods and material
The method section has been largely improved following the recommendations of the reviewer and I thank the authors for the clarifications they have made which serve the quality of the paper.
L. 169: remove “is” => Abies pindrow mostly occurred.
Results
I am still waiting for the improvement of some figures, which I am convinced would facilitate the reading.
L. 298-303: I am surprised by the youth of your trees, is it related to the local management?
L. 314-315: It's useless, it's already an interpretation and what's more you don't do climate reconstruction.
Fig. 4 and 5: I confirm that the addition of species names at the top of the graphs would facilitate reading and understanding on these two figures.
Discussion
Without wishing to be pushy, I still find the separation into two paragraphs not very useful. Also, I don't understand what paragraph 2 of section 4.1 adds. You can say that the statistical data confirm the relevance of using these data to study ring-climate relationships but stop talking about climate reconstruction, especially with chronologies of 40 to 70 years.
L. 497-499: These conclusions must be tempered; you are not studying the resilience or adaptive capacity of the species.
L. 504: No, your study only provides information on the impact on growth
L; 521-530: I am still not convinced by your management recommendations. I am more used to reading that we should favor garden forests where cohorts of different ages and species coexist. Clearcuts are largely proscribed to fight against climate change.
Conclusion:
The conclusion seems to me to be good as it is
Author Response
Dear Reviewer,
We highly appreciate (and enjoyed) your comments (2nd round), and our sincere thanks to you for your time and effort to help us to maintain the thematic coherence of the Introduction Section, in particular.
Thank you again for helping us to improve the manuscript.
Please see the attachment- Review Report
Sincerely
Neupane et al.

Reviewer 2 Report
Dear authors.
Thank you very much for the provided responses to my comments. The quality of your answers is very good. You answered to all my comments with a plausible way. The paper has significantly improved. However, I have some more minor comments. First, the added figure 1 is not good. The resolution is low. Also, the background is white!! Please, revise the figure. In the frame (pu and left) which shows Nepal, you should add the countries which have borders with Nepal. In the main figure (Dolakha district) you should add a satellite background and cities (please see the attached file). In lines 43-45, you say "...ecological drought, storms, and floods....". Please, rephrase like "...ecological drought, storms, floods, wildfires, soil erosion and forest dieback,...", and add the following studies (doi.org/10.3390/land11101705 and doi.org/10.1007/s00267-020-01363-9) with the reference [6]. Finally, make a final check for word errors and grammatical mistakes.

Author Response
Dear Reviewer,
We highly appreciated your comments and suggestions. We addressed all of the comments you have provided. Thank you again for helping us to improve the manuscript.
Please see the attachment.
Sincerely
Neupane et al.
